# RXR-Mediated Remodeling of Transcriptional and Chromatin Landscapes in APP Mouse Brain: Insights from Integrated Single-Cell RNA and ATAC Profiling

**DOI:** 10.3390/cells14241970

**Published:** 2025-12-11

**Authors:** Yi Lu, Xuebao Wang, Carolina Saibro-Girardi, Nicholas Francis Fitz, Radosveta Koldamova, Iliya Lefterov

**Affiliations:** Department of Environmental and Occupational Health, University of Pittsburgh, Pittsburgh, PA 15261, USA; yil197@pitt.edu (Y.L.); xuw61@pitt.edu (X.W.); cag237@pitt.edu (C.S.-G.); nffitz@pitt.edu (N.F.F.)

**Keywords:** Alzheimer’s disease, retinoid X receptor, bexarotene, single-cell transcriptomics, chromatin accessibility, transcription factor footprinting

## Abstract

**Highlights:**

**What are the main findings?**
RXR activation induces selective transcriptional rewiring in APP/PS1 mouse brain through a three-tiered regulatory cascade: direct RXR binding at accessible chromatin, activation of secondary transcription factors, and propagation through metabolic and inflammatory gene networks.Integration of scRNA-seq, snATAC-seq, and ChIP-seq reveals that baseline RXR/LXR dimerization dominates the control state, while Bexarotene treatment redirects regulatory influence toward stress-response pathways, upregulating lipid and sterol metabolism, proteostatic regulators, and developmental transcription factor programs.

**What are the implications of the main findings?**
The three-tiered regulatory architecture demonstrates how limited direct RXR binding events amplify into large-scale transcriptional programs, explaining RXR’s broad influence on neuroinflammation, lipid homeostasis, and synaptic function—processes disrupted in Alzheimer’s disease.The multi-modal analytical framework provides a generalizable atlas-building approach for mapping ligand-activated transcription factor networks in complex tissues, applicable beyond RXR to other nuclear receptors and context-dependent transcriptional regulators.

**Abstract:**

Ligand-activated Retinoid X Receptors (RXRs) regulate gene networks essential for neural development, neuroinflammation, and metabolism. Understanding how RXR activation influences chromatin architecture and gene expression may reveal mechanisms relevant to neurodegenerative diseases. We used Bexarotene-treated APP/PS1ΔE9 mice to study RXR-mediated regulatory mechanisms by integrating single-nucleus ATAC-seq (snATAC-seq) with single-cell RNA-seq (scRNA-seq) and validating differentially accessible chromatin peaks using RXR ChIP-seq. Transcription factor (TF) footprinting analysis mapped regulatory networks activated by ligand-bound RXR. Our integrated analyses revealed a multilayered transcriptional cascade initiated by RXR signaling. We identified RXR-centered regulatory circuits involving heterodimer activation, upregulation of downstream TFs, and induction of metabolic pathways relevant to neural function. Detailed analysis of neuronal TF networks revealed that Bexarotene modulates RXR’s role through existing regulatory scaffolds rather than creating new ones. This study demonstrates that combining scRNA-seq, snATAC-seq, and ChIP-seq enables comprehensive analysis of RXR-mediated transcriptional regulation. RXR activation orchestrates cell-type-specific chromatin remodeling of gene networks controlling neuroinflammation, lipid metabolism, and synaptic signaling, providing mechanistic insights into RXR-dependent transcriptional programs in Alzheimer’s disease pathology.

## 1. Introduction

Nuclear receptors (NRs) are ligand-activated transcription factors (TF) that regulate diverse metabolic pathways and cellular processes implicated in health and disease, including neurodevelopment, neuroinflammation, and neurodegeneration. Retinoid X Receptors (RXRs) occupy a central position within this family. Expressed as α, β, and γ isoforms in both humans and mice, RXRs function primarily as obligate heterodimer partners for more than 20 other nuclear receptors, mediating transcriptional regulation through DNA binding and recruitment of coregulators [1,2]. For decades, 9-cis RA—a stereoisomer of all-trans retinoic acid—was considered the endogenous ligand for RXRα, a view that evolved when R. Ruhl et al. (2015) identified 9-cis-13,14-dihydroretinoic acid as a physiologically relevant RXR ligand in mice [3]. Structural studies established the modular architecture of RXRs and revealed how ligand binding alters receptor conformation, enabling heterodimerization, DNA recognition, and transcriptional activation [4,5]. These properties place RXRs at the hub of gene regulatory networks that integrate metabolic and developmental cues [2,6].

In the nervous system, RXR signaling has been linked to processes ranging from neural differentiation to cholesterol metabolism and inflammatory regulation [7]. More than two decades ago, RXR activation was proposed as a therapeutic avenue in Alzheimer’s disease (AD), largely through its role in lipid handling and neuroinflammatory control [7]. ABCA1, an ATP-binding cassette transporter critical for cholesterol efflux, emerged early as an RXR-responsive gene and has since been implicated across multiple neurodegenerative contexts. Although the present study focuses on transcriptional and chromatin-level mechanisms, multiple independent investigations, including our own, have consistently documented reproducible phenotypic effects of Bexarotene, an RXR agonist in AD model mice. Research demonstrated that Bexarotene significantly improved cognition and memory in APP mice [8,9,10,11], although its effect on plaque load remains controversial [8,11,12]. Bexarotene’s increased *APOE* expression and protein level was observed following short-term treatment in APP/PS1 mice [8,13] and long-term treatment in 3xTg-AD mice [10]. Additional studies connect Bexarotene to regulating brain immune responses by reducing disease-associated microglia markers while enhancing Aβ phagocytosis [13,14], modulating pro-inflammatory profiles [15,16], and promoting neurodevelopmental gene expression patterns that regulate neuronal differentiation and hippocampal neurogenesis [17,18]. Moreover, a recent study in a non-AD model demonstrated that RXR activation modulates lipid metabolism and promotes lysosomal clearance of α-synuclein aggregates [19], providing further evidence for conserved RXR-mediated regulatory mechanisms across neurodegenerative conditions. Collectively, these findings lend independent support to the molecular framework identified here, even in the absence of additional behavioral or histological validation assays. These findings underscored the potential importance of RXR activity in brain homeostasis [11,13,15,17,18,19,20]. Yet, despite extensive studies of individual RXR target genes and pathways, the broader transcriptional and chromatin-level consequences of RXR activation in the brain remain poorly defined [6,21].

Recent advances in computational methods have enabled distinct but complementary approaches to study gene regulation from single-cell data [22]. The Seurat and Signac frameworks are widely used to integrate single-cell RNA sequencing (scRNA-seq) with single-nucleus ATAC sequencing (snATAC-seq), providing robust cell-type annotation, differential gene expression, and chromatin accessibility analysis at population and single-cell levels [23]. However, these pipelines primarily highlight accessible regions without directly inferring TF occupancy. In contrast, the TOBIAS suite of tools was specifically developed for ATAC-seq footprinting and allows high-resolution mapping of TF binding events, enabling reconstruction of regulatory networks that underlie observed chromatin dynamics [24]. By combining these approaches, we gain both a broad, cell-type-resolved view of transcriptomic and epigenomic remodeling and a mechanistic understanding of TF activity downstream of RXR activation.

We investigated how RXR activation remodels gene regulatory networks in the context of amyloid pathology. Using APP/PS1ΔE9 mice [25], we applied snATAC-seq, integrated with scRNA-seq and complemented by RXR ChIP-seq and TF footprinting [24,26]. This integrative strategy enabled us to link chromatin accessibility, RXR binding, and transcriptional output at cell-type resolution. Our analysis reveals that RXR activation engages heterodimer partners and induces multilayered transcriptional cascades across neuronal and glial populations. By constructing RXR-centered TF networks, we delineate how nuclear receptor signaling orchestrates cell-specific regulatory programs in the brain. These findings provide a mechanistic framework for understanding RXR function in neurodegenerative contexts and establish a resource for exploring nuclear receptor biology at single-cell resolution.

## 2. Materials and Methods

### 2.1. Animals and Treatment

APP/PS1ΔE9 [B6.Cg-Tg (APPswe, PSEN1dE9)85Dbo/Mmjax] transgenic mice (referred to as APP/PS1) were purchased from The Jackson Laboratory (USA). Heterozygous experimental animals were bred in-house and male and female mice were used for all experiments. All animal procedures were performed in accordance with the guidelines outlined in the Guide for the Care and Use of Laboratory Animals from the United States Department of Health and Human services and were approved by the University of Pittsburgh Institutional Animal Care and Use Committee. All experiments were conducted in compliance with the ARRIVE guidelines. All animals were littermates and housed with a 12 h light/dark cycle with ad libitum access to food and water. APP/PS1 mice aged 4 to 6 months were randomly assigned to either treatment group (100 mg/kg/day) or Control group (Vehicle treatment), with paired distribution according to age and sex. Only the lab technician was aware of the group allocation at all stages until data analysis. Bexarotene (Thermo Scientific Chemicals, Waltham, WA, USA; #J63701MC, CAS 153559-49-0) was prepared in a Vehicle solution of corn oil containing 1% DMSO on the day of treatment. The animals received 10 μL per gram of body weight of either Bexarotene or Vehicle solution by oral gavage for 10 consecutive days.

### 2.2. Perfusions and Brain Tissue Processing

Mice were anesthetized by I.P. injection of Avertin (1.25% tribromoethanol, 2.5% 2-methyl-2-butanol, 250 mg/kg body weight). Blood was collected via cardiac puncture of the right ventricle using EDTA-treated syringes, followed by transcardial perfusion with 20 mL of 0.1 M PBS, pH 7.4. The blood was centrifuged to collect plasma for storage. For single-cell assays, one hemisphere was dissected to remove the cerebellum, olfactory bulb, and subcortex. The other hemisphere was drop-fixed in 4% phosphate-buffered paraformaldehyde at 4 °C for 48 h before being transferred to 30% sucrose for storage.

### 2.3. Tissue Dissociation

Freshly dissected mouse brain tissue was immediately dissociated using the Adult Brain Dissociation Kit (Miltenyi, Gaithersburg, MD, USA; #130-107-677) following manufacturer’s instructions with minor modifications. Briefly, brain tissue was dissected into small pieces in Hank’s balanced salt solution (HBSS, Gibco, Grand Island, NY, USA; #14025092), and tissue pellets were incubated in enzyme mix with gentle rotation and mechanical dissociation. The resulting suspensions were filtered and washed. Next, tissue debris were removed using Debris Removal Solution (Miltenyi) following manufacturer’s instruction. Cell pellets were washed once using Dulbecco’s phosphate-buffered saline (DPBS) containing 0.04% bovine serum albumin (BSA) and filtered before cell counting. Cell viability was assessed using Trypan Blue, ranging from 85% to 95%.

### 2.4. Nuclei Isolation and Library Preparation for snATAC-Seq

Single-cell suspensions for scATAC-seq were obtained from two cohorts to ensure adequate sample size. The first cohort consisted of cryopreserved cells from the eight mice used for scRNA-seq (Section 2.3); after thawing, five samples (two Bexarotene and three control) passed quality control. A second cohort of five additional age-matched mice (three Bexarotene and two control) was treated identically and processed fresh, resulting in five samples per treatment group for scATAC-seq. All samples underwent the same nuclei isolation protocol described below.

Cryopreserved cells were thawed in 37 °C water bath, checked for viability, and incubated with 100 μL lysis buffer (10mM Tris-HCl pH 7.4, 10 mM NaCl, 3 mM MgCl_2_, 0.1% Tween-20, 0.1% IGEPAL, 0.01% Digitonin, and 1% BSA) for 3 min on ice, followed by a washing step (10 mM Tris-HCl pH 7.4, 10 mM NaCl, 3 mM MgCl_2_, 0.1% Tween-20, 1% BSA). Nuclei concentration was assessed using AOPI staining on Countess II (Thermo Fisher Scientific, Waltham, WA, USA) and adjusted to 1000 nuclei/μL. Droplet-based libraries were immediately prepared after nuclei isolation using the Chromium Next GEM Single Cell ATAC Kit v2 and ChIP H Single Cell Kit (10× Genomics, Pleasanton, CA, USA) according to manufacturer’s instructions. Single-nuclei suspensions were incubated with transposase for 30 min and used for GEM generation and barcoding on Chromium Controller (10× Genomics, Pleasanton, CA, USA) targeting at least 10,000 nuclei per sample. After cleanup, libraries were constructed and indexed using Single Index Kit N Set A (10× Genomics, Pleasanton, CA, USA). Bioanalyzer High Sensitivity DNA kit (Agilent) was used for quality control, and pooled libraries were sequenced by UPMC Genome Center (Pittsburgh, PA, USA) on Illumina NovaSeq S2 PE100 according to manufacturer’s recommendations.

### 2.5. Chromatin Immunoprecipitation and Sequencing–ChIP-Seq

Chromatin immunoprecipitation and sequencing (ChIP-seq) was performed according to our routine protocol and as before [17,18]. Briefly, brain lysates of mice treated with Bexarotene or Vehicle were sonicated with three pulses of 15 s at amplitude 30, followed by a 120 s pause and three pulses of 15 s at amplitude 40 using a Model 705 Sonic Dismembrator (Fisher Scientific, Pittsburgh, PA, USA), to obtain fragments of 200–600 bp. For immunoprecipitation, we used a rabbit polyclonal anti-RXR (ΔN 197, #sc-774, Santa Cruz, CA, USA). ChIP libraries were generated using TruSeq ChIP Sample Prep Kit (Illumina, San Diego, CA, USA) following manufacturer’s protocols. For each of the steps, samples were purified by AMPure XP beads (Beckman Coulter, Brea, CA, USA). Adapter-ligated samples were separated on a 2% agarose gel to obtain 250–300 bp size-range of DNA fragment to remove unligated adapters. The libraries were validated by Agilent Technologies 2100 Bioanalyzer (Agilent Technologies, Santa Clara, CA, USA) to check the size, purity, and concentration of the sample before the sequencing, and sequencing was performed on Illumina HiSeq2000 (Illumina, San Diego, CA, USA) instrument at the Functional Genomics Core, UPenn, Philadelphia (http://fgc.genomics.upenn.edu/, accessed on 22 November 2025).

### 2.6. Quantification and Statistical Analysis

#### 2.6.1. scRNA-Seq Data Analysis

Single cell RNA-seq data were generated as previously by our group ([27,28]) and were used in this study for integrative multi-omic analysis. These transcriptomic data were processed exactly as described in the original publication and were incorporated here to enable joint interpretation with newly generated ATAC-seq datasets. Re-analysis was limited to normalization and differential expression procedures necessary for cross-modal integration; all newly presented results arise from analyses that were not part of the original RNA-seq report.

Single-cell barcoded reads were demultiplexed and aligned to the mouse reference genome (GRCm38) using Cell Ranger pipeline v7.0 (10× Genomics). Cells with over 200 unique molecular identifiers (UMIs) were selected for single cell gene expression analysis. Each of the scRNA-seq libraries were read into R (v4.2.0) and processed using Seurat package [29,30] (v4.2.1) as previously described [28]. Cells were filtered by the following criteria: (1) unique feature counts >200 and <5000, (2) total counts < 50,000, and (3) <10% of mitochondrial gene counts. Potential doublets were identified and removed using the DoubletFinder [31] (v2.0) package. After filtering, 37,634 cells were kept and SCTransform (v2) was used for normalization and scaling. We performed principal component (P.C.) analysis followed by clustering using top 10 P.C.s at 0.25 resolution. Differential expression was performed using “MAST” algorithm [32] to identify cluster marker genes and help manually annotate cell types for each cluster. Cell types of interest were then subset as individual Seurat objects for further analysis. The mean expression of cell-type specific gene sets was added to the matrices using “AddModuleScore” function, and an additional cell filtering step removed subclusters or cells expressing non-selective gene sets. For differential expression between groups, we used “MAST” algorithm, and a gene was considered differentially expressed if it had a Bonferroni-corrected *p* value < 0.05.

#### 2.6.2. snATAC-Seq Data Processing and Cell Type Annotation

Raw sequencing reads were demultiplexed and mapped onto the mouse reference genome (GRCm38) using “cellranger-atac count” pipeline (10× Genomics). Arrow files were created from fragment files in ArchR (v1.0.1) [33]. High-quality nuclei were selected based on two criteria: (1) transcription start site (TSS) enrichment score greater than 4, and (2) number of mapped fragments between 1000 and 100,000. Doublets were identified using the “addDoubletScores” function and filtered with the “filterDoublets” function. Iterative LSI dimensionality reduction and clustering were performed using ArchR. Cluster identities were defined after unconstrained integration with scRNA-seq profile. Peaks were called using MACS2 [26] (v2.2.9.1). We used the Peak matrix to identify Peak-to-gene links through “addPeak2GeneLinks” function, enabling further correlation analysis between peak accessibility and gene expression.

#### 2.6.3. Identification of Candidate TF Regulators

ChromVar [34], implemented in ArchR package, was used to predict enrichment of TF activity on a per-cell basis. To identify functional TFs across each cell type, Pearson correlation was calculated between the motif enrichment Z-score and the ATAC-inferred gene expression using the “correlateMatrices” function (FDR < 0.01 and correlation > 0.5). In addition, Pearson correlation analysis was also performed between ATAC-inferred gene expression and gene expression level from RNA-seq profiles.

#### 2.6.4. Analysis of Differentially Accessible Peaks

The processed ArchR project was converted to Signac “Seurat Object” format using the ArchRtoSignac package [35] (v1.0.5) for differential accessibility analysis. Differentially accessible regions between the two treatment groups were determined by the “FindMarkers” function in the Signac package (v1.14.0), employing a logistic regression (L.R.) framework. Functional annotation of associated genes was performed using DAVID (v6.8, https://david.ncifcrf.gov, accessed on 3 October 2024).

#### 2.6.5. Transcription Factor Footprinting and Regulatory Networks

Accessible peak regions of each cell type from different samples were called using MACS2 (v2.2.9.1) from within Signac (Signac::CallPeaks) with default parameters. Following the standard TOBIAS (0.17.0) pipeline [24], we inferred TF activities through footprinting analysis on these differentially accessible peaks in both conditions–Bexarotene treated and Controls. The first step of the analysis is Tn5 insertion bias correction using the ATACorrect module. After correcting for the Tn5 insertion bias to highlight the effect of protein binding, ScoreBigWig was then used to calculate the footprint score based on the accessibility in flanking background regions and depth of the local footprint. Next, we applied the BINDetect module to estimate bound motif sites and differential motif scores between the two treatment groups based on footprint scores, sequence, and motifs across major cell types. For each TF, the global log2FC distribution was compared against a background distribution obtained by subsampling log2FC values from all motif sites, producing a differential binding score and an empirical *p*-value. TFs with scores in the top or bottom 5% or with −log10 (*p*-value) above the 95th percentile were considered significantly altered between Bexarotene-treated and Control conditions. To classify sites as bound or unbound, we compared each site’s footprint score to a background distribution generated from randomly sampled accessible regions; sites meeting significance threshold (*p* < 0.001) were labeled as bound. Finally, the TF–TF regulatory network was constructed using the CreateNetwork module by linking bound TFBSs within promoter regions to their target genes that encode transcription factors, thereby establishing regulatory edges from source TFs to the TFs they regulate. The resulting network was imported into Cytoscape (v3.10.3) [36] for visualization.

#### 2.6.6. ChIP-Seq Analysis and Validation of snATAC-Seq Data

We analyzed FASTQ files derived from sequencing libraries of Bexarotene (n = 2) and Vehicle (n = 2) treated APP/PS1 mice at the same age and treatment schedule as those used for scRNA-seq and snATAC-seq experiments. Reads were aligned to the mouse reference genome (GRCm38) using Bowtie2 (v2.5.4) with the local option. Low quality reads (mapping score < 2) were removed by Samtools (v1.21). After quality filtering, we obtained a total of 35.1 million uniquely mapped reads for the Bexarotene-treated group and 29.5 million for the Vehicle group, averaging approximately 16.1 million reads per sample. Peak calling was performed with MACS2 at a *q*-value threshold of 0.05, comparing Bexarotene vs. Vehicle groups. To validate genomic sites identified by TOBIAS BINDetect as enriched RXR motifs following Bexarotene treatment, we calculated the overlapping of those regions to the regions identified as enriched by ChIP-seq following Bexarotene treatment. Genomic coordinates of overlaps were determined by Bedtools (v2.31.1) with 10 kb genomic windows. The overlapping regions were annotated by Uropa with default options. The list of overlapping regions identified by ChIP-seq and ATAC-seq were further analyzed in Metascape [37] for multilist comparative analysis, integration of the annotation information, and enrichment analysis using GO processes, KEGG pathways, Reactome gene sets, canonical pathways, and CORUM complexes [38,39,40,41,42]. Overlapping peak sequences annotated to TF genes were scanned using MEME Suite FIMO (v5.5.7) with default settings to identify candidate RXRα heterodimer motifs.

All data processing using R and python packages and command line tasks were conducted on RHEL Server equipped with Intel Platinum 8352M CPU, 1TB of RAM, 128 cores.

## 3. Results

### 3.1. Single-Cell RNA and ATAC Profiling Identify Major Cell Types in APP/PS1 Mouse Brain

In this study (Figure 1A, showing study design and profiling strategy), we profiled the brains of 5 month-old APP/PS1 mice, an age characterized by early amyloid deposition and gliosis [25]. Mice were randomized to Vehicle treatment or RXR activation treatment by Bexarotene, and snATAC-seq libraries were generated. After quality control and integration with sc-RNA-seq data [27], we obtained chromatin accessibility profiles from 61,353 nuclei and transcriptional landscapes from 37,640 single cells.

Unsupervised clustering and UMAP projection identified major brain cell populations, including astrocytes, microglia, neurons, oligodendrocytes, endothelial cells, macrophages, pericytes/vascular smooth muscle cells, and a cluster containing choroid plexus epithelial and ependymal cells (Figure 1B, UMAP plots of integrated datasets). Cells from RXR-activated and Vehicle groups mapped to highly overlapping clusters, indicating that global cellular composition was preserved across conditions. The proportions of annotated peaks across major cell types are shown in Figure 1C, confirming balanced lineage representation across samples.

Marker gene analysis confirmed cluster identity (Figure 1D, heatmap of top marker genes by cell type). For example, *Aqp4* marked astrocytes, *Tmem119* microglia, *Slc17a7* excitatory neurons, *Cldn11* oligodendrocytes, *Pf4* macrophages, *Vtn* pericytes/VSMCs, and *Prlr* choroid plexus/ependymal cells. Chromatin accessibility analysis of cell type-specific marker genes paralleled expression patterns (Figure 1E, accessibility tracks for representative marker genes), further supporting robust annotation. Peak-to-gene linkage analysis highlighted correlated regions of chromatin accessibility and gene activity across distinct cell populations (Figure 1E, bottom), providing an initial framework for mapping RXR-dependent cis-regulatory interactions.

Taken together, these analyses establish a high-quality single-cell and single-nucleus atlas of the APP/PS1 mouse brain. All major neuronal and glial populations were robustly identified, with comparable representation between RXR-activated and Vehicle groups. The proportions of annotated peaks across cell types (Figure 1C) confirmed balanced lineage representation. This atlas provides a solid foundation for investigating RXR-dependent changes in chromatin accessibility and transcriptional regulation.

### 3.2. RXR Activation Remodels Chromatin Accessibility and Prioritizes Upstream Regulators

To systematically identify transcriptional regulators shaping the chromatin-accessibility landscape after RXR activation, we performed motif enrichment using chromVAR within ArchR. This quantified TF motif enrichment across major cell types (Figure 2A–C, motif enrichment across conditions and cell types). We then prioritized regulators by correlating motif enrichment with ATAC-inferred gene activity (gene scores) and with RNA expression, identifying 88 high-confidence candidates (Pearson > 0.5, FDR < 0.01; Appendix A, list of candidate TFs). Examples include cell-type-specific regulators such as *Pax6* (astrocytes), *Lef1* (endothelium), *Neurod6* (neurons), *Runx3* (macrophages), *Mafb* (microglia), and *Nfe2l3* (oligodendrocytes) (Figure 2D–F, representative TFs with matched motif/activity profiles).

### 3.3. Integration of scRNA-Seq and snATAC-Seq Data

We next integrated scRNA-seq and snATAC-seq datasets to map cell-type, condition-specific cis-regulatory elements (CREs), and their putative target genes, using feature linkages (significant correlations between an ATAC peak and nearby gene expression). Linkages were restricted to peaks within 100 kb of each TSS, using the union of peaks present in >2% of cells in at least one cell type. Most peaks showed no significant difference between RXR-activated and Vehicle groups across the five major cell types (n = 14,170; denoted “ns” in the heatmap), but 1605 peaks were more accessible with RXR activation and 1310 were enriched in Vehicle (Figure 3A and Appendix A, peak counts and differentially accessible peaks by cell type). Differentially accessible peaks were concentrated in microglia and endothelial clusters and less in astrocytes, oligodendrocytes and neurons (see Appendix A).

Peak–gene linkages connected accessibility changes to expression differences in the same cell type (see the distribution plot showing the effect size for ATAC-seq results on Appendix A). Microglia and endothelial cells demonstrated the highest number of differential peaks at a distance <100 kb (Appendix A). In microglia, there were 1752 differential peaks (856 upregulated and 816 downregulated). In this cell type, RXR activation increased chromatin accessibility affecting categories such as cell migration (*Ccr1* [43]), cell differentiation (*Dusp6* [44]), inflammatory response (*Cx3cr1* [45]), phosphorylation (*Bmp2k*, *Cbl* [46]), HDL remodeling (*Apoe*, *Lipg*, *Lipc*, *Lpl* [47]) (Figure 3C, left). Bexarotene also decreased chromatin accessibility in gene ontology categories such as chromatin remodeling transcription (*Nr3c1*, *Fkbp5*, both genes are specific for microglia and macrophages), cell differentiation (*Notch2*, *Spi1* [48] gene is microglia master regulator), and the phospholipase C-activating G protein-coupled receptor signaling pathway (*P2ry12* [45], *Lrp1* [49]) (Figure 3C, right). Peaks linked to *Apoe*/*Apoc* locus were significantly upregulated in microglia (*Apoe* log2FC = 1.66; adj-p = 0.028 and *Apoc1* log2FC = 1.13, adj-p = 6.3 × 10^−33^) with several promoter-proximal peaks showing higher accessibility under RXR activation (Figure 3B, locus-level coverage and linked peaks). *Apoe* is a known RXR target gene under LXR/RXR heterodimer control, highlighting the central role of RXR signaling in lipid regulation; its mRNA was significantly increased after Bexarotene treatment (log2FC = 0.42; adj-p = 3.23 × 10^−6^). *Nr3c1* gene codes for the glucocorticoid receptor and *Fkbp5* is associated with intracellular trafficking of heterooligomeric forms of steroid hormone receptors, suggesting RXRs’ role in regulating this pathway [50]. We identified two peaks linked to *Fkbp5* locus which were significantly downregulated in microglia (log2FC = −0.94; adj-p = 2.2 × 10^−5^), coinciding with mRNA downregulation (log2FC = −0.16; adj-p = 6.6 × 10^−6^) in these cells (Figure 3B, locus-level coverage and linked peaks and associated violin plot representing RNA level).

Endothelial cells showed 1717 differential peaks increasing chromatin accessibility and 928 decreasing it (Appendix A). This cell type demonstrated the strongest overlap between differential accessibility and transcriptional responses. Upregulated GO categories were regulation of transcription involved in different cell functions (*Hes1*, *Bach2*, *Id3*) (Figure 3D, left). *Hes1* (Figure 3B, locus-level coverage and linked peaks) is a key regulator of cell differentiation and proliferation, particularly in neural and stem cell development [51]. *Bach2* is important for immune cell function and *Id3* is involved in T cell development. Other upregulated functions were angiogenesis (*Bcam* [52]), cell migration (*Macf1* [53]), actin cytoskeleton organization (*S1pr1*, Sphingosine-1-phosphate receptor 1 is a endothelial specific gene [54]), and regulation of cell shape (*Palmd* is endothelial-specific gene associated with Angiogenesis and vascular immunity [54]). Figure 3B shows the differentially accessible peaks and the representative locus of *Hes1* gene in endothelial cells (log2FC = 0.70; adj-p = 2.73 × 10^−8^) and its mRNA upregulation (log2FC = 0.38; adj-p = 6.9 × 10^−29^). Downregulated in endothelial cells were chromatin remodeling (*Sik1* and *Sik3*, salt inducible kinases 1 and 3 [55]), apoptotic process (*Ltbp4* is a key regulator of TGFBs that controls TGF-beta activation by maintaining it in a latent state [56]), *Gpx4* (glutathione peroxidase [30]), *Pla2g4a* [57], and regulation of potassium ion transmembrane transport (*Edn3*, Endothelin 3 [58]) (Figure 3D, right). Figure 3B shows the differentially accessible peak of *Sik1* locus in endothelial cells (log2FC = −0.29; adj-p = 3.55 × 10^−8^) and its mRNA downregulation (log2FC = −0.14; adj-p = 1 × 10^−4^).

In astrocytes, there were 276 differential peaks (26 up- and 252 downregulating, Appendix A). Overall, gene ontology analysis of genes associated with differentially accessible peaks in astrocytes indicated enrichment of biological categories such as regulation of transcription (*Nupr1*), chromatin remodeling, brain development (Appendix A). *Nupr1* (Nuclear protein 1) is a transcription regulator associated with resistance to stress and is involved in numerous cell functions such as cell-cycle, apoptosis, autophagy, and DNA repair [59] (Figure 3B). Treatment increased chromatin accessibility near the *Nupr1* transcription start site (log2FC = 0.52; adj-p = 6.28 × 10^−4^) that was associated with increased expression (mRNA log2FC = 0.14; adj-p = 2.2 × 10^−9^).

In Oligodendrocytes there were 185 differential peaks (64 up- and 121 downregulating, Appendix A). Bexa treatment affected categories such as transmembrane receptor protein tyrosine kinase signaling (*Pdgfra* and *Mertk*) and nervous system development (*Zeb1* [60]) (Appendix A). *Pdgfra* [61] is an oligodendrocytes specific gene which was upregulated after treatment (mRNA log2FC = 0.27; adj-p = 0.021), matching the increased chromatin accessibility with a peak close to the promoter (log2FC = 0.72; adj-p = 4.9 × 10^−4^, see Figure 3B for coverage plots of differentially accessible peaks). *Mertk*, a tyrosine kinase that was reported upregulated by Bexarotene [62]) also showed increased expression after treatment in our study (mRNA log2FC = 0.11; adj-p = 0.03), accompanied by the increased chromatin accessibility with a peak close to the promoter (log2FC = 0.51; adj-p = 0.006). There were relatively a small number of differential peaks in Neurons (total number 110, see Appendix A). The GO terms were associated with dendrite morphogenesis, synaptic transmission, and regulation of synaptic plasticity (Appendix A).

We conclude that RXR activation drives cell-type-specific remodeling of chromatin accessibility, nominates upstream TF regulators with matched motif-expression relationships, and links differential accessibility to gene expression via peak–gene correlations. Microglia and endothelial cells show the strongest CRE remodeling, converging on inflammatory and angiogenic/developmental pathways, respectively.

### 3.4. Transcription Factor Footprinting Reveals RXR-Centered Regulatory Circuits

To explore how RXR activation alters transcriptional regulation, we analyzed open chromatin profiles from astrocytes, endothelial cells, oligodendrocytes, and microglia using TOBIAS, a computational footprinting framework for genome-wide TF binding dynamics [24] (Figure 4A). TOBIAS corrects for Tn5 insertion bias, identifies protected footprints, and quantifies motif occupancy, enabling detection of TF binding events within accessible chromatin regions.

Differential binding analysis (BINDetect) identified TFs with significant changes between RXR-activated and Vehicle groups (top and bottom 5% by binding score; Figure 4B, Appendix A, heatmap of significant TFs). Some TFs—including *Ets1*, *Erg*, and *Gabpa*—showed stronger binding in the Vehicle group across all four cell types, whereas *Neurod2* exhibited higher binding in endothelial cells, microglia, and oligodendrocytes. By contrast, TFs identified in astrocytes were primarily unique for these cells and displayed the most prominent cell type-specific responses.

Comparing footprinting changes with differential expression (Figure 4), 20–27% of TFs showed consistent regulation at both levels (Figure 4C, Appendix A). For example, *Olig2* had higher footprinting scores in astrocytes and oligodendrocytes (astro-foot log2FC = 0.02, oligo-foot log2FC = 0.01) and also showed higher gene expression (astro-DEG log2FC = 0.09, oligo-DEG log2FC = 0.13). Representative altered footprints included *Alx1* in astrocytes, *Erg* in endothelial cells, *Spi1* in microglia, and *Neurog1* in oligodendrocytes (Figure 4D).

Because RXR and its permissive heterodimers are the primary targets of ligand activation, many predicted RXR-responsive genes encoded TFs themselves. To connect these responses, we applied the TOBIAS network module and constructed directed TF regulatory networks (Figure 4E). This analysis predicted a cascade initiated by ligand-activated RXR, involving 25 primary TFs that collectively activate 141 additional TFs, revealing an RXR-centered regulatory hierarchy across brain cell types.

Footprinting analysis uncovered widespread RXR-dependent changes in TF binding, with astrocytes showing the strongest cell-specific responses. Integration with transcriptomics confirmed consistency between binding and expression for a subset of TFs, and network analysis revealed an RXR-initiated cascade that propagates through multiple downstream regulators.

The TOBIAS analysis thus identified not only differential TF binding events but also an RXR-initiated cascade of regulatory interactions. To further dissect how these cascades manifest in neuronal systems, we next applied TF network reconstruction using footprint-informed edges. This approach enabled us to visualize differential regulatory hierarchies and compare RXR-centered transcriptional networks across treatment groups at cell-type resolution.

### 3.5. Differential Regulatory Networks in Neuronal Systems

Given the central role of neuronal dysfunction in APP/PS1 pathology, we next examined TF regulatory networks reconstructed from footprinting results in neurons (Figure 4F). Network analysis compared RXR-activated and Vehicle groups, starting from differentially bound TFs identified by BINDetect (e.g., *Meis1*, *Atoh1*, *Neurod2* in Bexarotene RXR-activated group; *Foxn1* and *Rxra* in Vehicle-treated group). The resulting networks encompassed 134 TFs and thousands of edges, with comparable size, density (≈0.18–0.20), and global connectivity between conditions. This indicates that RXR activation does not dismantle the neuronal regulatory scaffold but instead modulates localized TF influence within an otherwise robust architecture.

Across both conditions, a consistent core of universal regulators—including *Klf4*, *Egr2*, *Zfx*, *Foxn1*, and *Nrf1*—emerged with the highest out-degrees, forming a stable, radiating subgraph irrespective of the initiating seed TF. These nodes thus represent foundational regulators of the neuronal TF landscape.

RXR activation selectively altered specific cascades and hubs. For example, a cascade from *Atoh1* to *Neurod2* was enhanced, and *Neurod2* itself emerged as a prominent regulatory hub. In contrast, the control network emphasized broad cascade initiators such as *Foxn1* and key integrators like RXRα. Thus, differential TF binding translated into shifts in how regulatory signals were initiated, propagated, and integrated within neuronal networks.

Neuronal TF networks remain structurally robust under RXR activation but display localized rewiring, with selective enhancement of cascades (e.g., *Atoh1* → *Neurod2*) and shifts in hub influence. These changes provide mechanistic insight into how RXR signaling reshapes transcriptional regulation in neurons without disrupting core network architecture.

The TF network analysis therefore highlights how RXR activation alters neuronal regulatory hierarchies through selective rewiring while maintaining overall structural stability. To independently validate these chromatin- and network-based predictions, we next performed RXR ChIP-seq, focusing on whether footprint-derived regulatory elements correspond to direct RXR binding sites.

### 3.6. ChIP-Seq Validation of RXR-Bound Regulatory Elements

To validate RXR binding predicted from chromatin accessibility, we performed ChIP-seq with an RXR antibody (Details in Methods) in APP/PS1 mouse brain tissue, using the same treatment paradigm as for single-cell assays. Peak calling identified 11,957 significant RXR-binding sites in RXR-activated versus Vehicle samples (Figure 5B). Of these, 3908 overlapped with enriched ATAC-seq peaks (Figure 5A). Among the 1251 overlapping peaks, 777 were located at transcription start sites (TSSs) and another 60 within upstream promoter regions, indicating that nearly 70% of overlapping peaks represent proximal cis-regulatory elements (CREs) likely contributing to gene regulation under RXR activation (Figure 5C).

Integration of ATAC-seq motif enrichment and RXR ChIP-seq binding revealed complex overlap among gene lists (Figure 5D, Circos plot of ChIP–ATAC motif concordance). GO and pathway analysis of the 1251 overlapping peaks highlighted enrichment in diverse cellular processes (Figure 5E). Many of the predicted target genes clustered in KEGG pathways associated with neurodegenerative disorders, including Alzheimer’s disease (AD), Parkinson’s disease (PD), Huntington’s disease (HD), prion disease (PrD), and amyotrophic lateral sclerosis (ALS). Differential expression analysis of these neurodegeneration-related genes across astrocytes, oligodendrocytes, endothelial cells, and microglia confirmed altered expression consistent with the footprinting predictions (Figure 5G).

These results demonstrate that integrating ATAC-seq footprinting, RXR ChIP-seq, and scRNA-seq provides direct evidence linking RXR occupancy at CREs to changes in gene expression. Importantly, not all genes in the enriched networks are direct RXR targets—consistent with the notion that RXR activation propagates through TF cascades (see also Figure 4). Nevertheless, the overlap of RXR binding with differentially accessible regions and disease-related pathways validates RXR as a central regulator of chromatin remodeling and transcriptional programs in the APP/PS1 brain.

RXR ChIP-seq confirmed that a large fraction of differentially accessible regions identified by ATAC-seq correspond to direct RXR binding sites, many of them proximal to gene promoters. Integration with RNA-seq and footprinting linked these sites to expression changes, including genes involved in neurodegenerative disease pathways, strengthening the evidence for RXR as a key regulator of brain transcriptional networks.

The ChIP-seq validation thus confirmed that many of the accessibility changes identified by ATAC-seq correspond to direct RXR binding sites, including those linked to neurodegeneration-related pathways. To place these findings into a broader framework, we next integrated scRNA-seq, snATAC-seq, and ChIP-seq data. This integrative analysis enabled us to assign RXR binding to candidate cis-regulatory elements and connect them to cell-type-specific transcriptional programs across the APP/PS1 brain.

### 3.7. Integrative Single-Cell Analysis Links RXR Activation to Candidate Cis-Regulatory Elements

To further integrate accessibility and binding data, we analyzed transcriptional regulators with overlapping ATAC-seq and RXR ChIP-seq peaks within ENCODE regulatory signatures. This approach identified seven TFs of particular interest—*Myc*, *Arid5a*, *Creb3l2*, *Egr1*, *Zic2*, *Gli2*, and *Nfatc2*—each showing cell-type-specific expression and regulatory associations (Figure 5G, lower panel).

Each TF exhibited a distinct regulatory architecture reflecting its functional specialization. Arid5a was associated with pro-inflammatory regulators such as STAT3, NF-κB, and AP-1, consistent with immune pathways [63,64]. *Creb3l2* linked to ATF4, XBP1, and CEBPβ, supporting a role in ER stress and acute-phase responses [65,66]. *Egr1* was connected to SRF, ELK1, and CREB, aligning with activity-dependent transcription and neuronal plasticity [67,68]. Developmental TFs such as OTX2, SOX2, and LHX2 were linked to *Zic2*, while *Gli2* was connected to Hedgehog pathway mediators GLI1, SUFU, and FOXC1 [69,70]. None of the above TFs has been reported as transcriptionally controlled by RXR.

Among the seven, only *Myc* and *Nfatc2* showed previously recognized direct regulatory input from retinoid receptors, specifically RXR/RAR heterodimers. Our ChIP-seq analysis and integrative framework provide new evidence for RXR involvement in their regulation. *Myc*, classically controlled by MAX and SP1, displayed evidence for functional interaction with RXR. *Nfatc2*, a key immune regulator, has been shown to physically interact with RXRα and RARα, enhancing retinoic acid response element binding. In our data, RXR/Nfatc2 co-binding was significantly altered by RXR activation (Appendix A), underscoring its responsiveness to retinoid signaling [71,72].

These analyses confirm that RXR acts not only through direct DNA binding but also via co-regulatory partnerships with other TFs, extending its influence into developmental, immune, and stress-response networks.

Integrative analysis of ATAC-seq, ChIP-seq, and RNA expression identified RXR occupancy at cis-regulatory elements linked to *Myc*, *Nfatc2*, *Egr1*, and other key TFs. These findings validate RXR’s role as both a direct DNA-binding factor and a co-regulatory partner, reinforcing its central position in transcriptional networks that control neuronal, immune, and developmental responses.

## 4. Discussion

Our study demonstrates that RXR activation remodels the chromatin and transcriptional landscape of the APP/PS1 mouse brain through cell-type-specific regulatory cascades. The analysis highlights both conserved network architecture and selective rewiring of TF hierarchies, particularly in neuronal and glial systems. Notably, RXR/NR1H3 (LXR) dimerization, long recognized as a fundamental mechanism, emerged as a key regulatory axis in the control state, suggesting that baseline RXR activity may be driven by endogenous ligands. RXR activation shifted this balance, redirecting regulatory influence toward other TF pathways. Consistent with established RXR biology, the transcriptional programs we observed following Bexarotene treatment prominently affected lipid and sterol metabolism—such as upregulation of *Msmo1*, *Insig1*, *Hmgcr*, *Hmgcs1*, *Cyp51*, and *Idi1*—as well as genes involved in lipid transport and homeostasis (*Lrp1*, *Apod*). In parallel, increased expression of stress-response and proteostatic regulators (*Hspa5*, *Hspa8, Hsp90ab1*) and developmental or state-transition TFs (Sox, Hes, *Ccnd* family) suggests that RXR activation enhances metabolic support, modulates glial activation programs, and may improve resilience to inflammatory or injury-related stress in AD-like pathology. These RXR-driven pathways closely align with known roles of RXR/LXR signaling in maintaining brain lipid balance, limiting neuroinflammation, and supporting neuronal function. Importantly, the effects on APOE and APOE-HDL in the brain and ISF and CSF of human subjects were independently validated in multiple studies [8,15,73,74,75]. These findings place RXR at the intersection of lipid metabolism, immune regulation, and neuronal plasticity, central processes that are disrupted in Alzheimer’s disease.

Beyond the specific biological results, our work illustrates the power of integrating scRNA-seq, snATAC-seq, and ChIP-seq to reconstruct TF regulatory networks at single-cell resolution. This multi-layered approach bridges transcriptional and epigenomic measurements, enabling validation of predicted TF–DNA interactions and revealing how linear receptor signaling can generate complex regulatory cascades. Although RXR is a nuclear receptor, the framework presented here is broadly applicable to other TFs, provided their activation context—whether physiological, developmental, or pharmacological—is well defined. Although the number of biological replicates in the snATAC-seq and ChIP-seq datasets is modest, the analyses rely on cross-modal concordance rather than single-modality peak calls. Differential accessibility, TF footprinting, and ChIP-seq binding were interpreted only when supported by multiple evidence layers and aggregated across thousands of nuclei, which substantially mitigates biological variability and reduces the likelihood of false-positive signals.

The modest overlap between RXR ChIP-seq peaks and ATAC-accessible regions is consistent with expectations for comparing bulk binding maps to single-nucleus accessibility profiles. Many ChIP-seq-exclusive peaks likely reflect binding events in cell populations not captured or not accessible in the snATAC-seq dataset, as well as transient or cooperative interactions that do not produce stable accessibility changes. The intersecting peaks therefore represent the most conservative set of high-confidence direct RXR targets.

Our integrative analyses also show how RXR operates through (a) direct DNA binding/**primary effects**, corresponding to direct RXR binding at accessible chromatin regions validated by ChIP-seq; (b) through partnerships or **secondary effects**, reflected in RXR-dependent activation of TFs such as *Nfatc2*, *Creb3l2*, and *Myc* that themselves exhibit altered footprinting and expression; and finally (c) **tertiary downstream remodeling**, comprising broader metabolic, inflammatory, and developmental gene-expression programs propagated by these second-layer TFs with other TFs, such as NFATc2, SP1, and CREB family members. This co-regulatory architecture explains how a relatively small set of RXR binding events can propagate through large-scale transcriptional networks, extending RXR’s influence into immune responses, stress adaptation, and developmental programs. Such findings underscore the value of building an “atlas” of RXR-centered TF networks across brain cell types as a resource for future studies of neurodegeneration.

There are several limitations of the present study. First, we acknowledge that while TOBIAS provides robust, bias-corrected predictions of differential occupancy, footprinting-based regulatory TF networks depend on motif accessibility and do not, by themselves, establish biochemical causality. Furthermore, we did not directly link RXR-dependent changes in TF networks to behavioral or inflammatory phenotypes in APP/PS1 mice. Numerous prior studies, including from our own laboratory, have reported that RXR agonists improve cognition and modulate inflammation in AD models, but replicating those endpoints was beyond the scope of this integrative genomic analysis. Future studies may incorporate behavioral assessments to establish functional relevance of our chromatin accessibility findings. Key proposed directions include integrating longitudinal cognitive testing with multi-omics profiling to correlate chromatin accessibility changes with behavioral outcomes, cell-type-specific functional validation of neuroinflammatory and synaptic effects, RXR isoform-specific genetic studies, and replication across additional AD models and human tissue to determine whether these chromatin changes translate to therapeutically relevant cognitive improvements. Second, the APP/PS1ΔE9 model, while widely used and well-characterized for studying amyloid pathology, primarily recapitulates the amyloid-centric aspects of Alzheimer’s disease and does not fully exhibit tau pathology, extensive neurodegeneration, or the full spectrum of pathological features observed in human AD. Validation in additional AD models, including those with tau pathology, would be valuable to establish whether RXR-mediated chromatin remodeling represents a broadly applicable therapeutic mechanism across different aspects of AD pathogenesis. Furthermore, while Bexarotene shows high RXR selectivity, potential effects mediated through RXR heterodimerization with other nuclear receptors should be considered when interpreting our results. Another limitation is that, given the scale of the data, we could not exhaustively explore all RXR heterodimers and TF co-occurrences. A more comprehensive analysis of RXR–TF partnerships across cell types will be the subject of future work.

## 5. Conclusions

By integrating scRNA-seq, snATAC-seq, and RXR ChIP-seq, we constructed a cell-type-resolved regulatory framework for RXR signaling in the brain. RXR activation induces selective rewiring of transcriptional networks while preserving overall regulatory architecture, demonstrating how nuclear receptor signaling translates into coordinated transcriptional cascades. These findings provide mechanistic insight into the role of RXR in neuronal and glial systems and establish a broadly applicable framework for studying ligand-activated TFs in complex tissues.

## Figures and Tables

**Figure 1 cells-14-01970-f001:**
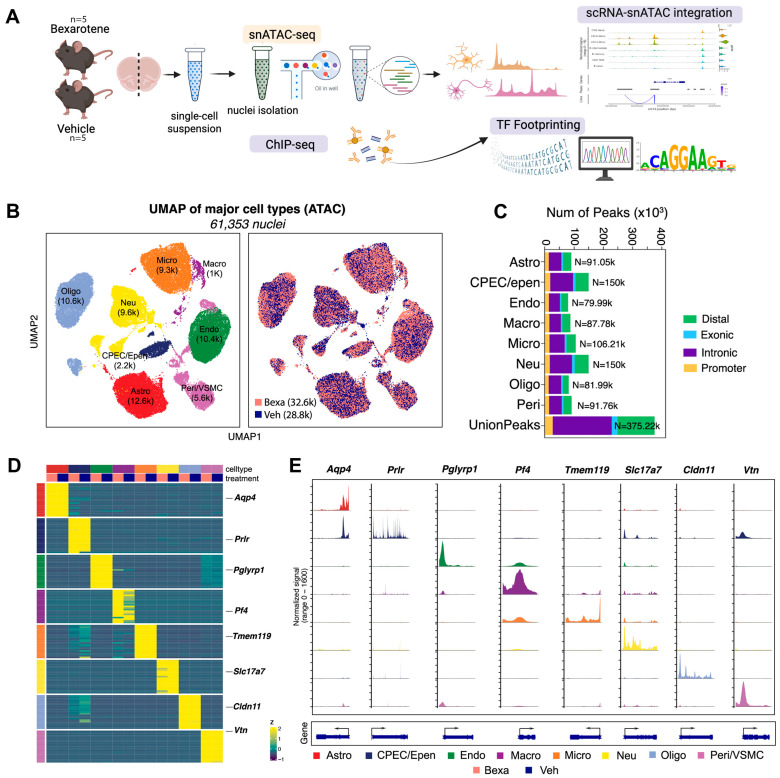
(**A**) Experimental design. APP/PS1 mice (5 months old, male and female) were treated with RXR agonist (Bexa; 100 mg/kg/day, oral) or Vehicle (corn oil with 1% DMSO) for 10 days. Brains were dissociated into single-cell suspensions and processed for snATAC-seq (Bexa, n = 5; Veh, n = 5) library preparation using the 10x Genomics Chromium platform. (**B**) UMAP projections showing sample treatment distribution across integrated scRNA-seq and snATAC-seq datasets. Major cell types are annotated as Astro (astrocytes), CPEC/Epen (choroid plexus epithelial/ependymal cells), Endo (endothelial cells), Macro (macrophages), Micro (microglia), Neu (neurons), Oligo (oligodendrocytes), and Peri/VSMC (pericytes/vascular smooth muscle cells). (**C**) Proportions of annotated peak distributions across major cell-type groups (n = 5 per group). (**D**) Heatmap showing normalized expression of top marker genes for each cell type. (**E**) Coverage plots showing chromatin accessibility across major cell types for representative marker genes.

**Figure 2 cells-14-01970-f002:**
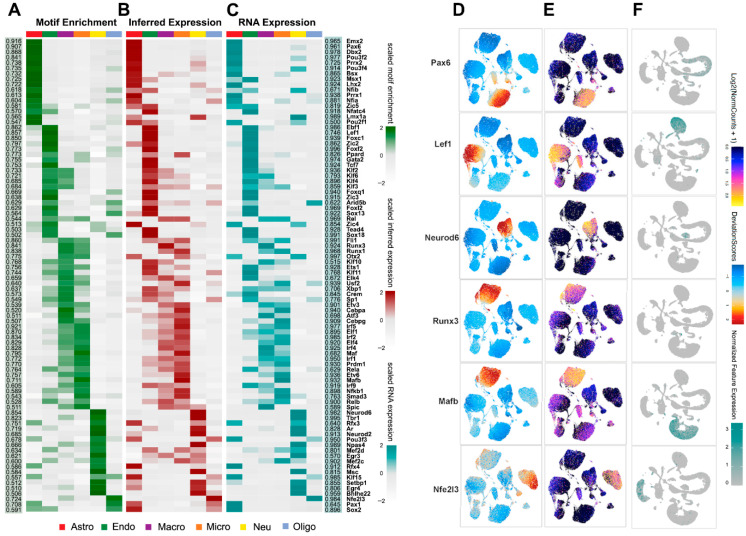
(**A**–**C**) Heatmaps of 88 candidate transcription factor (TF) regulators across major cell types shown as chromVar motif enrichment (**A**), inferred gene expression (**B**), ATAC gene-score from ArchR), and RNA expression level ((**C**), from scRNA-seq). The two side panels show Pearson correlations between ATAC-inferred TF expression and chromVar (**left**, highlighted in green) or gene expression (**right**, highlighted in cyan). (**D**–**F**) UMAP embeddings of example candidate TF regulators as of motif enrichment (**D**) and inferred gene expression (**E**) from ATAC-seq (ArchR), as well as gene expression (**F**) from RNA profile (Seurat). The examples of cell-specific TF regulators are *Pax6* in astrocytes, *Lef1* in endothelial cells, *Neurod6* in neurons, *Runx3* in macrophages, *Mafb* in microglia, and *Nfe2l3* in oligodendrocytes.

**Figure 3 cells-14-01970-f003:**
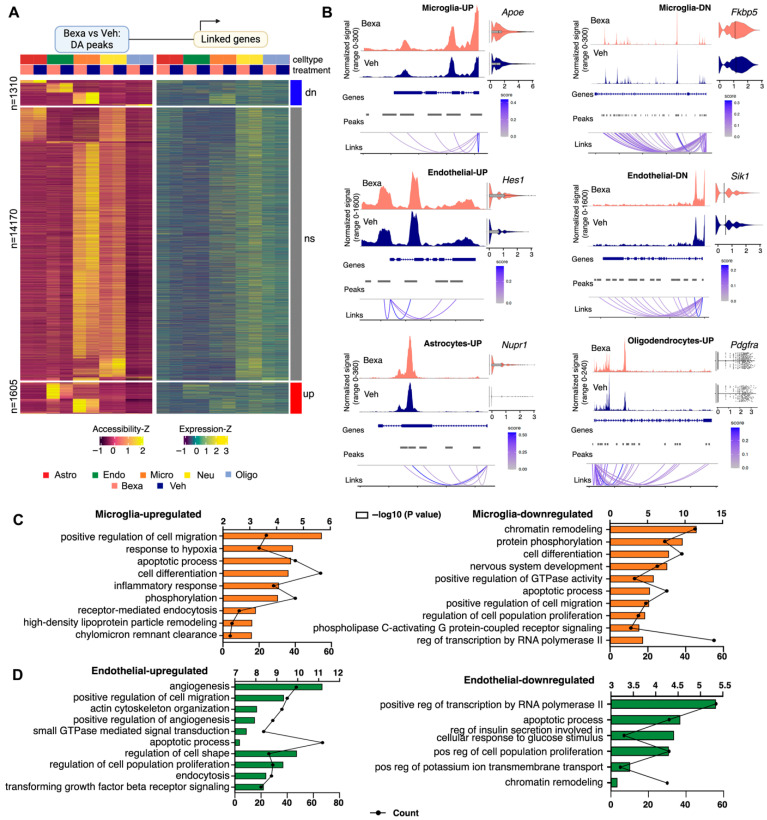
(**A**) Heatmap of differentially accessible peaks (**left**) and their linked genes (**right**) between RXR-activated and Vehicle-treated groups across major cell types. Peaks are categorized as downregulated (dn, n = 1310), non-significant (ns, n = 14,170), or upregulated (up, n = 1685) based on Bexarotene vs. Vehicle comparison (adjusted *p*-value < 0.05). Samples are grouped by cell type and treatment. Z-scores are shown for both accessibility and expression levels. (**B**) Coverage plots of differentially accessible peaks in representative loci, including gene annotations, peak coordinates, genomic links, and expression levels under each condition. Shown are representative peaks at adjusted *p* values < 0.05 and DEG. Details in the text of the results. (**C**,**D**) GO enrichment analysis of biological processes associated with more accessible peaks and upregulated genes (**left**) or less accessible peaks and downregulated peaks (**right**) in microglia (**C**) and endothelial cells (**D**) following RXR activation.

**Figure 4 cells-14-01970-f004:**
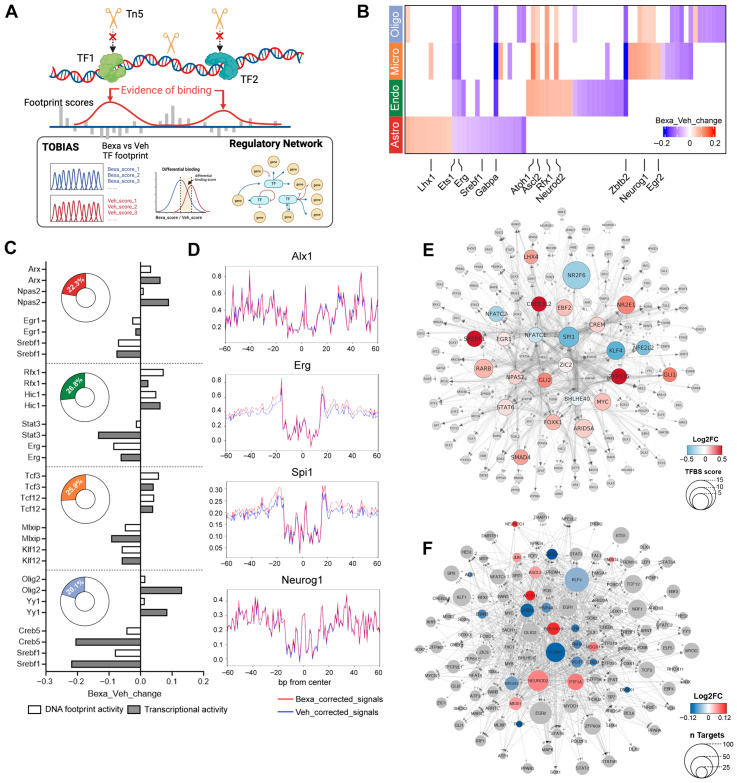
(**A**) Schematic workflow for TF footprinting and regulatory network construction. (**B**) Heatmap of differentially bound TFs between RXR-activated and Vehicle groups in astrocytes, endothelial cells, microglia, and oligodendrocytes. Red = higher binding under RXR activation; blue = higher in Vehicle. (**C**) Bar charts showing representative TF occupancy (white) and gene expression changes (gray; log2FC, RXR versus Vehicle). Donut plots show the percentage of TFs with concordant occupancy and expression changes. (**D**) Representative TF footprints with significantly altered occupancy (FDR < 0.05): Alx1 in astrocytes (Bexa_Veh_change = 0.04, pval = 6.93 × 10^−53^), Erg in endothelial cells (Bexa_Veh_change = −0.09, pval = 1.84 × 10^−108^), Spi1 in microglia (Bexa_Veh_change = −0.12, pval = 1.61 × 10^−142^), and Neurog1 in oligodendrocytes (Bexa_Veh_change = 0.05, pval = 9.59 × 10^−47^). Red = corrected signals in RXR-activated; blue = Vehicle. (**E**) RXR-centered TF regulatory network. Nodes represent TFs bound in TF promoters associated with RXR. Node shapes denote hierarchical levels: circles = direct RXR targets; octagons = indirect TFs downstream of direct targets. For circles, node color indicates mean log2FC in binding score (RXR versus Vehicle); node size represents TFBS score. (**F**) Neuronal TF regulatory network showing TFs with significantly different binding scores (RXR versus Vehicle). Colored circles = significantly altered TFs; gray circles = direct TF targets of these TFs. Circle size indicates the number of direct TF targets; color represents mean log2FC in binding score.

**Figure 5 cells-14-01970-f005:**
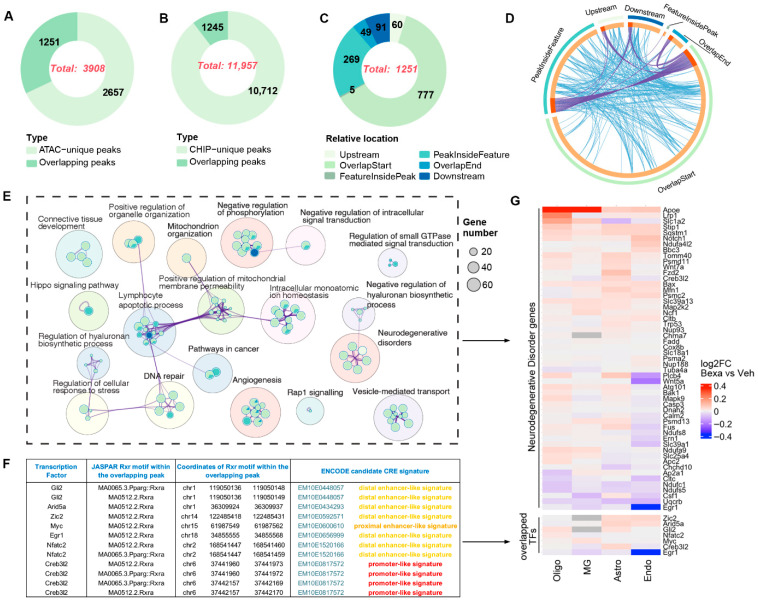
(**A**–**C**) Number of unique and overlapping peaks identified by ATAC-seq and ChIP-seq and their relative positions with respect to nearest gene features. (**D**) Circos plot visualizing overlap among gene lists from RXR ChIP-seq binding and ATAC-seq motif enrichment. Outer arcs = gene lists; inner arcs = member genes. Dark orange = shared genes; light orange = unique genes. Purple lines link genes present in multiple lists; blue lines link genes under the same significantly enriched ontology term. (**E**) Network of enriched pathways associated with genes overlapping ATAC-seq and ChIP-seq peaks. Dot size = number of enriched genes; sector area = proportion of overlapping genes in each category. GO and pathway enrichment were performed with Metascape and visualized with Cytoscape. (**F**) Table of TF names, JASPAR RXRα (PPARγ::RXRα) motifs, and coordinates within overlapping ATAC-seq/ChIP-seq peaks and ENCODE cCRE signatures. Because ChIP-seq was performed with an RXR antibody, only RXR homo- and heterodimer motifs are included. (**G**) Heatmap of log2FC expression for genes near overlapping ATAC-seq and ChIP-seq peaks that are implicated in neurodegenerative disorders. Lower panel: cell-type-specific expression of TFs located near overlapping peaks.

## Data Availability

Single nucleus ATAC-seq and ChIP-seq data have been deposited at GEO and will be publicly available under series accession number GSE283906 and GSE289039 upon publication. Single cell RNA-seq data [27] used in this study will be publicly available under series accession number [GSE283905]. These data were re-analyzed solely for integrative purposes in combination with the ATAC-seq datasets generated here. The current manuscript presents new analyses, new data modalities, and new biological conclusions that were not part of the original RNA-seq study.

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
