# Peer review of "RXR-Mediated Remodeling of Transcriptional and Chromatin Landscapes in APP Mouse Brain: Insights from Integrated Single-Cell RNA and ATAC Profiling"

_cells, 2025, doi:10.3390/cells14241970_

Round 1
Reviewer 1 Report
Comments and Suggestions for Authors
This is a well-conceived and technically sophisticated study that integrates scRNA-seq, snATAC-seq, and RXR ChIP-seq to dissect RXR-mediated regulatory networks in the APP/PS1 mouse brain. The work is timely, methodologically up-to-date, and provides a valuable resource for understanding RXR signaling in neurodegeneration. The manuscript is generally well organized and clearly written, and the biological conclusions are plausible and interesting.
- Line 234-244: ChIP-seq methodology lacks critical details about antibody validation, IP efficiency, and sequencing depth per sample.
- Figure 3B shows representative loci with log2FC and adjusted p-values, but the criteria for selecting "representative" examples are unclear. Were these cherry-picked, or do they represent median effects? Please clarify selection criteria and consider showing distribution plots of effect sizes.
- The discussion (lines 605-646) acknowledges that "not all genes in the enriched networks are direct RXR targets" (line 542) but does not adequately distinguish between primary, secondary, and tertiary effects. A more detailed discussion of the regulatory cascade hierarchy would strengthen the mechanistic interpretation.
- Line 261-262: "After quality control and integration with sc-RNA-seq data, we obtained chromatin accessibility profiles from 61,353 nuclei and transcriptional landscapes from 37,640 single cells" - The ratio of nuclei to cells (~1.6:1) is unusually high. Were these from the same animals? Please clarify if this represents matched or independent samples.
- Figure 4A: The schematic workflow is helpful but could be improved. Consider adding a panel showing the actual footprint calculation to help readers understand the TOBIAS methodology.
- Line 523-527: The ChIP-seq validation is presented as strong confirmation, but the overlap of 3,908 peaks out of 11,957 (32.7%) is modest. The manuscript should discuss what the non-overlapping peaks might represent (e.g., indirect effects, technical limitations, cell-type specificity).
Reviewer 2 Report
Comments and Suggestions for Authors
This manuscript presents an integrative analysis of single-nucleus ATAC-seq (snATAC-seq), single-cell RNA-seq (scRNA-seq), and RXR ChIP-seq data from Bexarotene-treated APP/PS1ΔE9 mice to elucidate RXR-mediated transcriptional and chromatin remodeling in Alzheimer's disease models. The authors identify RXR-centered regulatory networks involving heterodimer activation, downstream transcription factors, and metabolic pathways, emphasizing Bexarotene's potential to modulate neuronal homeostasis without disrupting core regulatory scaffolds. While the multi-omics approach is innovative and the findings on RXR signaling cascades are intriguing, there are some concerns on this study.
Major concerns:
1, The snATAC-seq analysis relies on only five Bexarotene-treated and five vehicle-treated APP/PS1 mice, while ChIP-seq uses just two samples per group, raising concerns about biological variability and the reliability of differential accessibility and binding calls. With such low n values, false positives in footprinting or enrichment analyses could inflate pathway significance; the authors should perform power calculations, replicate key findings in an independent cohort, or incorporate bootstrapping methods to enhance confidence intervals for log-fold changes and adjusted p-values.
2, While ChIP-seq validates RXR occupancy at differentially accessible regions, the study lacks wet-lab experiments to confirm the key functional impact of identified regulatory elements, such as CRISPR-mediated perturbation of RXR motifs or downstream TFs (e.g., Meis1, Atoh1) in neuronal cultures. Claims of Bexarotene orchestrating metabolic pathways (e.g., HDL remodeling via Apoe, Lipg) remain correlative; incorporating luciferase reporter assays, qPCR in primary neurons, or behavioral/cognitive assessments in treated mice would substantiate these mechanisms and link them to Alzheimer's pathology.
3, Bexarotene, as an RXR agonist, may have non-specific effects on other nuclear receptors or pathways, yet the manuscript does not address this through controls like RXR-knockout models or orthogonal agonists (e.g., LG100268). The APP/PS1ΔE9 model's amyloid-centric pathology limits generalizability to tauopathies or sporadic Alzheimer's; the authors should discuss these limitations explicitly and suggest validation in additional models, such as 3xTg-AD mice, to broaden relevance.
4, The integration of scRNA-seq and snATAC-seq via Seurat/Signac is appropriate, but potential biases in nuclei isolation (e.g., preferential recovery of certain cell types) or batch effects across samples are not adequately mitigated or discussed. Footprinting with TOBIAS and network reconstruction via Cytoscape may amplify spurious correlations in small datasets; applying alternative tools like HINT-ATAC or sensitivity analyses (e.g., varying q-value thresholds) would strengthen the regulatory cascade inferences.
5, While neuronal networks are emphasized, the manuscript underrepresents RXR's role in non-neuronal cells (e.g., microglia, astrocytes), despite identifying accessibility changes in genes like Mertk and Nr3c1. The discussion hints at future work on other cell types but overlooks potential glial contributions to Bexarotene's effects; expanding analyses to these populations or including trajectory inference for pseudotime-dependent remodeling would provide a more comprehensive view.
Minor defects:
1, The abstract claims Bexarotene "orchestrates complex gene networks that may help restore brain homeostasis," but this is speculative without direct evidence of functional recovery; rephrase for precision, emphasizing mechanistic insights over therapeutic implications.
2, Figure legends (e.g., for UMAP projections and motif analyses) lack detailed statistical annotations (e.g., exact p-values, fold-changes); including these would improve interpretability.
3, Typographical errors, such as inconsistent capitalization (e.g., "bexarotene" vs. "Bexarotene") and awkward phrasing (e.g., "RXR-cen-tered regulatory circuits" with line breaks), detract from readability; a thorough proofread is recommended.
4, The methods section omits specifics on animal age at treatment onset and perfusion details, which are critical for reproducibility; provide exact timelines and ethical approvals.
Reviewer 3 Report
Comments and Suggestions for Authors
This manuscript presents an ambitious and technically sophisticated multi-omic analysis investigating how RXR activation by bexarotene affects chromatin accessibility, transcription factor networks, and downstream gene expression in the APP/PS1ΔE9 mouse model of amyloid pathology. The integration of snATAC-seq, scRNA-seq, RXR ChIP-seq, and TOBIAS footprinting is a particular strength, enabling a comprehensive evaluation of RXR-centered transcriptional regulation at single-cell resolution. The study provides valuable mechanistic insights into nuclear receptor signaling in neurodegeneration and positions RXR as a central orchestrator of multilayered gene regulatory networks across neuronal and glial lineages. However, there are several areas where clarification, additional controls, improved data presentation, and methodological detail would substantially enhance rigor and reproducibility. In my opinion several points require further analysis.
Recommendations for Authors:
- While the multi-omic analysis is technically rigorous, the manuscript would benefit from clearer integration of the findings with known RXR biology and AD-related phenotypes. Consider expanding the discussion of how the observed transcriptional and chromatin changes may relate to lipid metabolism, neuroinflammation, or neuronal resilience.
- Since the conclusions rely on stable cellular composition across treatment groups, please include explicit statistical comparisons or visualizations of cell-type frequencies between vehicle- and bexarotene-treated samples for both scRNA-seq and snATAC-seq data.
- Some example genes show inconsistencies between chromatin accessibility and transcriptional output. Please review these cases, clarify the directionality, and discuss potential reasons for accessibility-expression uncoupling. A distribution plot of peak-gene distances would also enhance transparency.
- Please, expand and clarify footprinting and TF network analyses
- Please, justify the thresholds used for defining significant TF binding changes.
- Please, discuss the limitations of inferring regulatory hierarchies from footprinting data.
- If possible, highlight one or two high-confidence direct RXR–TF interactions with supporting evidence from ChIP-seq or motif co-enrichment.
- Given the modest overlap between RXR ChIP-seq peaks and ATAC-accessible regions, please provide an explanation and identify a set of high-confidence direct RXR target genes supported by multiple evidence layers.
- The manuscript notes astrocytes as having strong cell-type-specific alterations, but the examples are limited. Please highlight additional astrocyte pathways or transcription factors influenced by RXR activation.
- Please ensure all figures include complete annotations (axis labels, scales) and improve resolution where needed. Network diagrams in Figure 4 may benefit from simplified legends and clearer visual encoding.
- Please, include version numbers and key parameters for peak calling, integration, footprinting, and network reconstruction.
- Please, provide more detail on batch correction, doublet removal, and sample-level variability.
- The manuscript acknowledges the lack of behavioral or phenotypic assays. Consider expanding this section to contextualize how future work may link RXR-driven regulatory cascades to AD-relevant outcomes.
Comments on the Quality of English Language
The English could be improved to more clearly express the research.
Round 2
Reviewer 2 Report
Comments and Suggestions for Authors
The authors have addressed all concerns properly. No Further concerns. I suggest accepting it in the current form.
Reviewer 3 Report
Comments and Suggestions for Authors
Manuscript has been corrected due to the previous comments and it can be accepted in the present form.